# Molecular Dynamics Simulations of Docetaxel Adsorption on Graphene Quantum Dots Surface Modified by PEG-b-PLA Copolymers

**DOI:** 10.3390/nano12060926

**Published:** 2022-03-11

**Authors:** Mehdi Yoosefian, Mitra Fouladi, Leonard Ionut Atanase

**Affiliations:** 1Department of Chemistry, Graduate University of Advanced Technology, Kerman 7631885356, Iran; 2Department of Nanotechnology, Graduate University of Advanced Technology, Kerman 7631885356, Iran; mitra_foladi66@yahoo.com; 3Faculty of Medical Dentistry, Apollonia University of Iasi, 700511 Iasi, Romania

**Keywords:** graphene quantum dots, drug delivery, docetaxel, PEG-b-PLA copolymer, surface modification, drug solubility

## Abstract

Cancer is associated with a high level of morbidity and mortality, and has a significant economic burden on health care systems around the world in almost all countries due to poor living and nutritional conditions. In recent years, with the development of nanomaterials, research into the drug delivery system has become a new field of cancer treatment. With increasing interest, much research has been obtained on carbon-based nanomaterials (CBNs); however, their use has been limited, due to their impact on human health and the environment. The scientific community has turned its research efforts towards developing new methods of producing CBN. In this work, by utilizing theoretical methods, including molecular dynamics simulation, graphene quantum dots (GQD) oxide was selected as a carbon-based nanocarriers, and the efficiency and loading of the anticancer drug docetaxel (DTX) onto GQD oxide surfaces in the presence and in the absence of a PEG-b-PLA copolymer, as a surface modifier, were investigated. According to the results and analyzes performed (total energy, potential energy, and RMSD), it can be seen that the two systems have good stability. In addition, it was determined that the presence of the copolymer at the interface of GQD oxide delays the adsorption of the drug at first; but then, in time, both the DTX adsorption and solubility are increased.

## 1. Introduction

Today, cancer is treated by surgery, radiotherapy, and chemotherapy. Chemotherapy involves the administration of drugs that kill the cancer cells, preventing them from growing into other cells [1,2,3]. However, many people are afraid of chemotherapy because of its side effects, without knowing that these side effects can be controlled by the means of nanotechnology. Recently, the development of nanomedicine has revolutionized the treatment and cure of cancer by targeting accurate diagnoses, efficient and specific treatments, and real-time monitoring [4,5,6,7]. Moreover, some novel nanocarriers have been designed to optimize their physicochemical properties, such as size, softness, shape, surface charge, and modification, to achieve highly efficient delivery, to improve theranostic efficacy, and to reduce systemic toxicity [8,9,10,11].

The purpose of designing drug delivery systems based on nanocarriers is to overcome the defects and disadvantages of conventional drug formulations, reduce the frequency of drug use, increase the effect of the drug by focusing on the desired location, and reduce the amount of drugs required and provide controlled and sustained drug delivery [12]. The modern drug delivery system is the delivery of a drug at a specific time and at a controlled dose to specific drug targets; this is dramatically safer and much more effective than drug delivery throughout the body. Furthermore, new drug delivery reduces side effects and consumes lower doses.

One of the efficient methods to avoid the side effects of chemotherapy is to create new drug delivery systems which have the ability to encapsulate anticancer drugs inside nanocarriers and deliver them exclusively to tumor sites by active targeting [13,14,15,16]. Targeted delivery can also ensure that the drug reaches the area where it is needed, without any damage that may be transmitted through body systems, such as the gastrointestinal tract or the circulatory system [17]. In the design and development of drug delivery systems (DDSs), the goal is to achieve a system with proper drug loading efficiency and optimal release properties, characterized by a long half-life and low toxicity, and choosing a suitable administration route [18,19,20]. The most important factor in cancer treatment is the design of non-toxic and biodegradable nanocarriers for efficient anticancer drug delivery [21,22,23]. Recent developments in nanotechnology and polymer engineering have created a new and more efficient world for the treatment of cancer using nanoformulations. Carriers used in drug delivery include micelles, liposomes, nanoparticles, dendrites, liquid crystals, hydrogels, conjugates, cobosomes, and hexosomes [24,25,26].

Carbon nanoparticles are widely used in various applications [27,28,29]. Carbon nanotubes, graphene derivatives, and fullerene have attracted much attention [30,31,32,33,34]. Graphene has a honeycomb structure composed of SP^2^ single plates of carbon atoms with an electron arrangement. GQD can interact with biomolecules, so they can be used for drug and gene delivery by making appropriate surface changes [35].

Docetaxel (DTX) was approved by the FDA in 2004 as a highly potent chemotherapy drug. DTX is a taxoid antineoplastic agent used in the treatment of various cancers, such as locally advanced or metastatic breast cancer, gastric adenocarcinoma, metastatic prostate cancer, and head and neck cancer [36,37]. DTX can be used alone or in combination with other chemotherapy drugs. DTX is one of the most effective chemotherapeutic drugs for the treatment of cancer; however, it has serious side effects, including severe toxicity (including bone marrow suppression), nausea, hypersensitivity, peripheral neuropathy, loss of appetite, and musculoskeletal disorders [38,39,40]. In this study, GQD oxide was used to transport DTX, which was possible due to the adsorption process by the interaction between the DTX drug and GQD oxide. After this, the surface of the GQD oxide was modified by the PEG-b-PLA copolymer, and the ability of the modified surface of carbon quantum dot as a new DDS for DTX was investigated.

## 2. Materials and Methods

A series of MD simulations were performed on the anticancer drug DTX with quantum dot graphene oxide. All parameters of bonded and non-bonded interactions were selected from the Charmm36 force field, and the PDB and ITP of the DTX drug were performed based on this force field. Molecular dynamics simulation calculations were performed using GROMACS version 5.1.4 (Royal Institute of Technology and Uppsala University, Stockholm, Sweden), and VMD was used to view the entire simulation process [41,42]. Figure 1 demonstrates the chemical structure of the GQD oxide (C_365_O_22_H_48_), the PEG-PLA copolymer, and the DTX molecule.

Simulated cubic boxes were used to test systems and to solve the relevant structure in the TIP3P water model. In one system, the DTX and the GQD oxide nanocarrier were examined and, in the other system, the DTX and the GQD oxide nanocarrier in the presence of the PEG-b-PLA copolymer were examined. In order to achieve that, we inserted 10 polymer chains, containing 7 PEG repeat units and 3 PLA repeat units, into the simulated box. The systems were then minimized and balanced. The algorithm of “steepest” was used for integration in 50,000 steps, in order to perform simulations to minimize energy. Periodic boundary conditions were applied to eliminate the surface effect and keep its value constant. Initially, two equilibration steps in NVT and NPT ensembles were run for 100 ps for the temperature and pressure to reach their constant values. In both simulation systems, the temperature at 300 K and pressure at 1 bar were kept constant using the V-rescale thermostat and the Parrinello–Rahman barostat, respectively. The van der Waals interactions were modeled using a cut-off distance of 1.4 nm, and the long-distance electrostatic interactions were investigated using PME. Finally, the equations of motion were integrated using the leap-frog algorithm, with a time step of 2 femtoseconds and a total time of 50 nanoseconds.

## 3. Results and Discussion

Adsorption of the DTX anticancer drug on the GQD oxide surface in the presence and absence of PEG-b-PLA copolymer as a surface modifier was investigated using molecular dynamics simulations. For this purpose, descriptors, such as the total energy, the potential energy, the root mean square displacement, the radial distribution function, the contact area, and the number of hydrogen bonds, were considered.

### DTX Adsorption onto GQD Oxide

The behaviors of the DTX adsorption process on the GQD oxide nanocarrier in two different conditions are investigated in this section. In the first step, which was called DTX/GQD, the DTX is adsorbed on the GQD oxide. In the second step, which we call DTX/POL/GQD, the DTX is loaded on the GQD oxide in the presence of the PEG-b-PLA copolymer. Several snapshots at different time frames, corresponding to the presence and absence of the PEG-b-PLA copolymer in the simulation box, are given in Figure 2 and Figure 3, respectively.

In the DTX/GQD system, the drug is adsorbed by the nanocarrier from the beginning of the process. On the contrary, in the DTX/POL/GQD system, the presence of the copolymer causes instability in the system and initially prevents the drug from reaching the GQD oxide surface due to steric effect; but, over time, the drug is adsorbed onto the nanocarrier.

Different parameters, such as total energy and potential energy, were used as indicators to evaluate the stability of the system. The goal of minimizing energy is to bring the system to the most stable state possible. Using this operation, any spatial congestion or improper overlap in the system is eliminated, and the simulation can be started after the systems have reached the minimum energy. The total energy and potential energy for both the DTX/GQD and DTX/POL/GQD systems can be seen in Figure 4.

The results of the energies diagram show that the total and potential energy of the DTX/POL/GQD system is higher than that of the DTX/GQD system. This fact can be attributed to the spatial inhibition of the copolymer. Moreover, the results of potential diagrams show that both systems are stable. It is noteworthy that the energies are almost constant with increasing simulation time. Although the presence of polymers initially reduces the stability of the system, it ultimately leads to an increase in drug solubility.

The analysis of radial distribution function data shows, on average, how the particles in a system are arranged radially. Using RDF, one can study the interactions between different atoms and how they are placed next to each other. Atomic RDF can be useful at the nanometric level to better understand the molecular orientation of adsorbed drugs. RDF, or pair correlation function *g_AB_*(*r*), between particles of type A and B is defined in the following way:(1)$$g_{AB}(r)\;=\;\frac{\langle \rho_{B}(r)\rangle }{{\langle \rho_{B}\rangle }_{local}}$$
with 〈*ρ_B_*(*r*)〉 as the particle density of type B at a distance *r* around particles A, and 〈*ρ_B_*〉*_local_* as the particle density of type B averaged over all spheres around particles A with radius *r*_max_. Usually, the first peak in each diagram of the radial distribution function represents the first correlation between the species in the investigated system, and the distances between the two species of this correlation can be obtained from the peak location (see Figure 5). The position of the first peak for the drug and nanocarrier in the DTX/POL/GQD system is 0.806 nm, and the position of the first peak for the drug and nanocarrier in the DTX/GQD system is 0.726 nm.

The root mean square deviation (RMSD) was already calculated to check the convergence of the simulation. However, it can also be used for further analysis. The RMSD is a measure relating two structures. If the RMSD calculated for each combination of structures in a trajectory file, then it can be seen if there are groups of structures belonging together, sharing structural features. Such structures will have lower RMSD values within the group, and higher RMSD values with other structures. Representing the RMSD values in a matrix (Figure 6) also helps to identify transitions.

The RMSD analyses for both systems, as shown in Figure 6a, are fixed at approximately 0.5 ns. This means that both systems balanced quickly, with RMSD values averaging 5 nm, which indicates the stability of the systems during the 50 ns MD simulation.

In order to investigate the interaction between the drug molecule and the nanocarrier in both the DTX/POL/GQD and DTX/GQD systems, the relative distance between them was measured and is presented in Figure 6b. The center of mass distance between the drug molecule and the nanocarrier in the DTX/POL/GQD system is separated by an average of 2.8 nm in less than 3 ns, and, finally, it does not appear that the distance between the drug center and the nanocarrier has changed significantly. Thus, after about 5 ns of simulation, it becomes constant and equal to 0.2 nm; for the drug molecule and nanocarrier distance in the DTX/GQD system, it is constant from the beginning and equal to 0.2 nm. In the DTX/POL/GQD system, due to the spatial barrier of the copolymer, significant fluctuations can be observed. At first, the drug molecule and the nanocarrier were separated from each other; over time, the distance decreases, leading to the conclusion that the copolymer does not have an important effect on drug adsorption to the nanocarrier.

To better understand the interactions between the drug and the nanocarrier, the number of contacts between them was analyzed. As shown in Figure 6c, for the DTX/GQD system, the drug reaches the appropriate orientation after about 5 ns and then remains at the highest constant value until the end of the simulation. However, for the DTX/POL/GQD system, there is no contact between the drug and the GQD surface for the first 5 ns and then the drug approaches the GQD surface after it has passed through the copolymer and has reached a maximum contact of about 1000. Obviously, due to the presence of the PEG-b-PLA copolymer, the contact between the drug and the surface in the DTX/POL/GQD system is less than in the DTX/GQD system.

The Lennard-Jones energy is an approximate potential for describing the interactions between two atoms or molecules: Two neutral molecules feel both attractive and repulsive forces based on their relative proximity and polarizability. The sum of these forces gives rise to the Lennard-Jones potential. According to the results obtained from Figure 6d, it can be stated that the energy of Lennard-Jones between the DTX and GQD oxide in the DTX/POL/GQD system is lower than that in the DTX/GQD system, due to the spatial barrier provided by the PEG-b-PLA copolymer and van der Waals interactions.

## 4. Conclusions

In this study, the details of the interaction between the DTX and GQD oxide as a nanocarrier in the presence and the absence of the PEG-b-PLA copolymer were investigated by molecular dynamics simulations. In the design and development of drug delivery systems, the goal is to achieve a system with proper drug loading and the desired release properties required, along with a long half-life, low toxicity, and increased solubility. The most important factor is to recognize the most stable state of a drug in a biological system. After calculating and analyzing the relative energy of both systems according to the calculation of total energy, potential energy, and RMSD, it was determined that the two systems have good stability. Analysis of the distance between the drug molecule and GQD oxide showed that significant fluctuations are observed in the DTX/POL/GQD system due to the spatial arrangement of the PEG-b-PLA copolymer. The number of contacts between the drug and the nanocarrier in the DTX/GQD system increased. With an increasing number of drug and nanocarrier contacts in both systems, Lennard-Jones energy significantly decreased. Finally, RDF analysis shows that the peak of the drug mass center with a nanocarrier was higher in the DTX/GQD system, which indicates the intensity of the DTX with GQD oxide interaction in the absence of the copolymer, but had little effect on drug uptake on the nanocarrier surface. Finally, the presence of the copolymer increased drug solubility, which is a very important feature for anticancer drugs with low solubility.

## Figures and Tables

**Figure 1 nanomaterials-12-00926-f001:**
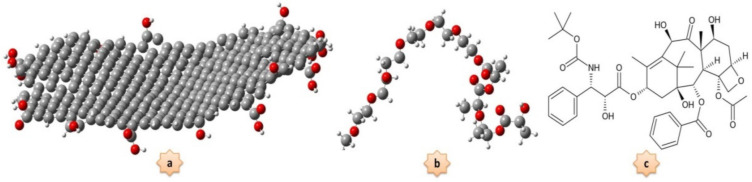
The chemical structure of (**a**) graphene quantum dot oxide; (**b**) PEG-PLA copolymer in the covalent radius with atom colors: white—hydrogen, gray—carbon, red—oxygen; and (**c**) docetaxel drug.

**Figure 2 nanomaterials-12-00926-f002:**
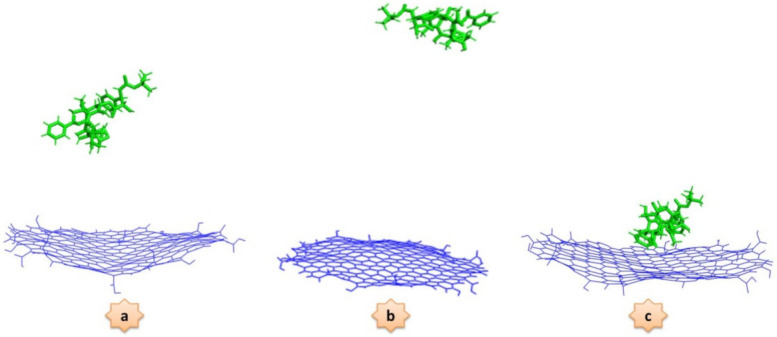
Representative snapshots corresponding to the interaction of docetaxel molecules with graphene quantum dot oxide at three different times: (**a**) 0 ns, (**b**) 25 ns, and (**c**) 50 ns.

**Figure 3 nanomaterials-12-00926-f003:**
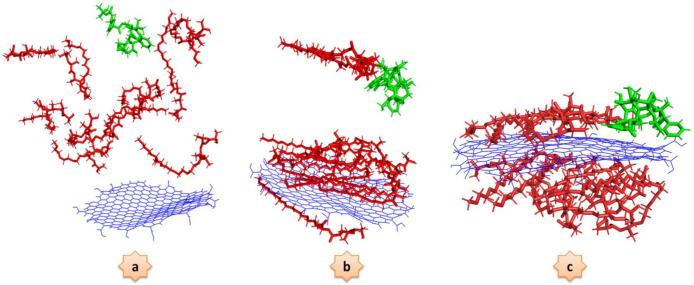
Representative snapshots corresponding to the interaction of docetaxel molecules and graphene quantum dot oxide in the presence of PEG-PLA copolymer at three different times: (**a**) 0 ns, (**b**) 25 ns, and (**c**) 50 ns.

**Figure 4 nanomaterials-12-00926-f004:**
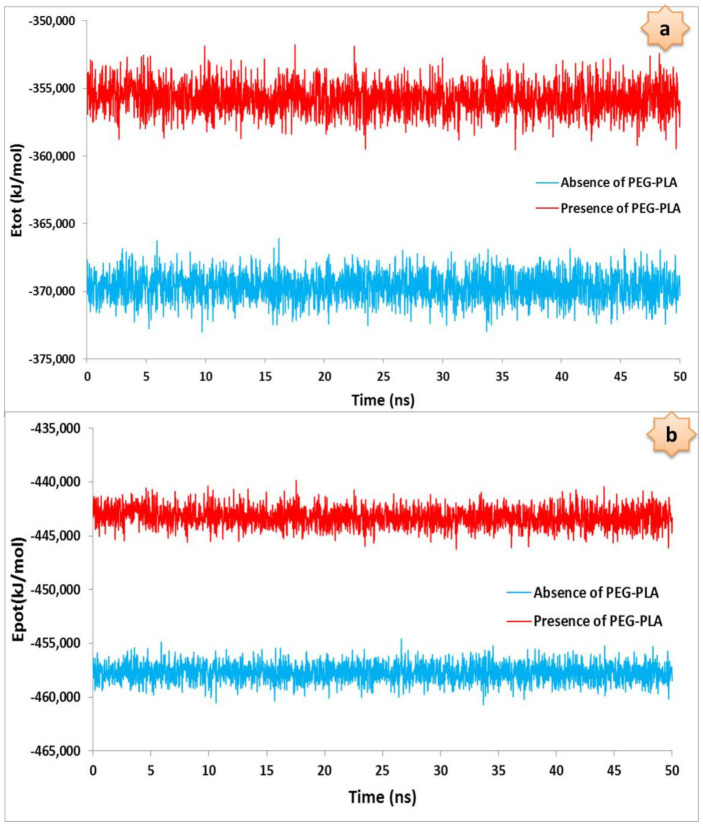
Total (**a**) and potential (**b**) energies for the docetaxel anticancer drug loading onto graphene quantum dot oxide in the presence and absence of the PEG-PLA copolymer.

**Figure 5 nanomaterials-12-00926-f005:**
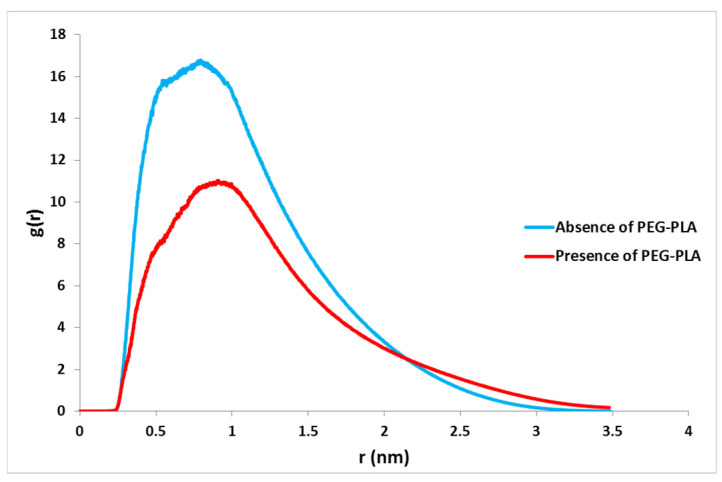
RDF of the docetaxel anticancer drug loading onto graphene quantum dot oxide in the presence and absence of the PEG-PLA copolymer over 50 ns MD trajectories.

**Figure 6 nanomaterials-12-00926-f006:**
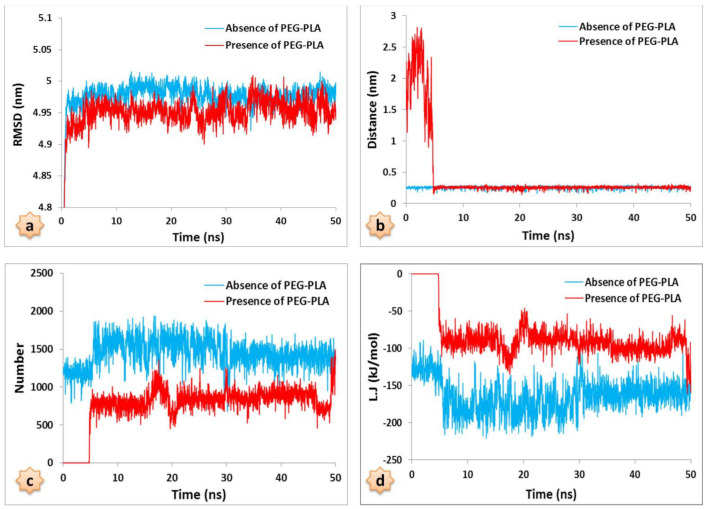
Trajectory analysis of the docetaxel anticancer drug loading onto graphene quantum dot oxide in terms of RMSD (**a**), distance (**b**), number of contacts (**c**), and L.J energies (**d**) in the presence and absence of the PEG-PLA copolymer over 50 ns MD simulation.

## Data Availability

Not applicable.

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
