# Peer review of "Molecular Dynamics Simulations of Docetaxel Adsorption on Graphene Quantum Dots Surface Modified by PEG-b-PLA Copolymers"

_nanomaterials, 2022, doi:10.3390/nano12060926_

Round 1
Reviewer 1 Report
This manuscript reports a Molecular Dynamics (MD) simulation of the interaction between docetaxel (DTX), an anticancer drug, and a graphene nanocarrier (GQD) with and without a block copolymer (PEG-b-PLA) that may enhance the system solubility in an aqueous environment. First of all, I should note that the general purpose of this paper is not fully clear to me. Maybe it only aims to show that the DTX interaction with GQD is favorable on the isolated nanocarrier, and also in the presence of the block copolymer. However, another issue is very important: the drug must be eventually released from the nanocarrier to be effective. This issue is not considered here: some words of comment about this point should however be given, in my opinion. Apart from that, there are a number of questions and comments/suggestions that the authors should consider before the manuscript could be accepted for publication. I list them here below.
- Which is the detailed structure of the graphene quantum dot? What are the peripheral groups shown in Fig. 1? And apart from that, graphene has no other modification: is it correct? And why is it mentioned later at line 99 and in the following text as a GQD oxide nanocarrier? Please provide also a line drawing of DTX showing its chemical structure: Figure 1b is hardly significant.
- In Fig. 1, panel (a) shows the graphene nanocarrier, and (b) the DTX drug: please correct the caption.
- What is the SPCE216 water model? What does 216 stand for? And what is the molar mass or the degree of polymerization of the PEG and PLA blocks? And how many such blocks are present in the polymer?
- Line 90: what is meant by balanced? Equilibrated maybe? And how? Line 91: what does it mean "maintain the amount"? Line 93: what is the comfort level? Line 94: Kelvin should be indicated with a capital letter. Line 97: most likely, the Leap-Frog algorithm is meant.
- Please check Fig. 2: DTX appears to first increase its distance from GQD, and then to approach it (same side or the opposite one thanks to the periodic boundary conditions): is it correct?
- Fig. 4 is a bit unclear to me, in that it does not show any change along the trajectory. Maybe the systems were fully equilibrated through previous runs, but if it is so, it must be clearly stated.
- In Fig. 5, what is exactly r? Maybe the distance between the center-of-mass of DTX and the surface, or of the closest atom?
- Fig.6c and 6d: what is happening in the presence of PEG-b-PLA an the last 2-3 ns of the trajectory? Maybe the drug is eventually displacing part of the polymer to be in full contact with the graphene surface? I feel that longer runs would have been helpful.
- Line 192: Lennard-Jones, not Leonard-Jones. Line 193: Pauli's exclusion principle has nothing to do with the Lennard-Jones energy or the van der Waals interactions, here. Line 198: the presence of π-π interactions should be shown in more detail by an appropriate figure.
Author Response
- Which is the detailed structure of the graphene quantum dot? What are the peripheral groups shown in Fig. 1? And apart from that, graphene has no other modification: is it correct? And why is it mentioned later at line 99 and in the following text as a GQD oxide nanocarrier? Please provide also a line drawing of DTX showing its chemical structure: Figure 1b is hardly significant.
ANSWER: Chemical formula of the graphene quantum dot is C365O22H48. The peripheral groups have shown in Fig. 1 are hydroxy and carboxyl groups. Yes it is correct: graphene oxide has no other modification. Actually this is a graphene quantum dot oxide (GQD oxid), and in whole of manuscript corrected. A line drawing of DTX showing its chemical structure provided in Fig 1c.
- In Fig. 1, panel (a) shows the graphene nanocarrier, and (b) the DTX drug: please correct the caption.
ANSWER: The caption of Fig 1 corrected.
- What is the SPCE216 water model? What does 216 stand for? And what is the molar mass or the degree of polymerization of the PEG and PLA blocks? And how many such blocks are present in the polymer?
ANSWER: SPCE216 is a gro file in GROMACS program that used as solvent structure file that is incorrectly written as a water model. In fact, the water model used in this study was TIP3P, which was corrected. In the simulated box we insert 10 polymer chains containing 7 PEG blocks and 3 and PLA blocks.
- Line 90: what is meant by balanced? Equilibrated maybe? And how? Line 91: what does it mean "maintain the amount"? Line 93: what is the comfort level? Line 94: Kelvin should be indicated with a capital letter. Line 97: most likely, the Leap-Frog algorithm is meant.
ANSWER: Line 90: balanced means Equilibrated here. Minimized and balanced the systems have been done by the algorithm of “steepest” which was used for integration in 50,000 steps in order to perform simulations to minimize energy. Line 91: this sentence revised as: Periodic boundary conditions were applied to eliminate the surface effect and keep its value constant. Line 93: the simulation details of NVT and NPT revised completely. Line 94: Kelvin indicated with a capital letter. Line 97: Leap-Frog algorithm replaced.
- Please check Fig. 2: DTX appears to first increase its distance from GQD, and then to approach it (same side or the opposite one thanks to the periodic boundary conditions): is it correct?
ANSWER: Yes, it is correct and as you mentioned correctly, because of periodic boundary conditions it looks like this.
- Fig. 4 is a bit unclear to me, in that it does not show any change along the trajectory. Maybe the systems were fully equilibrated through previous runs, but if it is so, it must be clearly stated.
ANSWER: As you know, the potential energy depends on the number of atoms and the spatial effects between them. And as stated in the manuscript, in fact, low changes in energies show the stability of system during simulation. Also, as you mentioned, in minimization energy, NVT and NPT equilibrium steps system completely optimized.
- In Fig. 5, what is exactly r? Maybe the distance between the center-of-mass of DTX and the surface, or of the closest atom?
ANSWER: In Fig. 5, r is the distance between the center-of-mass of DTX and the GQD oxide surface.
- Fig.6c and 6d: what is happening in the presence of PEG-b-PLA an the last 2-3 ns of the trajectory? Maybe the drug is eventually displacing part of the polymer to be in full contact with the graphene surface? I feel that longer runs would have been helpful.
ANSWER: Fig.6c and 6d: Your opinion is completely correct and as shown in Fig 3c, the DTX is eventually displacing part of the polymer to be in full contact with the GQD oxide surface.
- Line 192: Lennard-Jones, not Leonard-Jones. Line 193: Pauli's exclusion principle has nothing to do with the Lennard-Jones energy or the van der Waals interactions, here. Line 198: the presence of π-π interactions should be shown in more detail by an appropriate figure.
ANSWER: Line 192: Lennard-Jones replaced. Line 193: Pauli's exclusion principle removed. Line 198: π-π interactions changed to van der Waals interactions.
Reviewer 2 Report
In this paper, the authors studied the efficiency and loading of anticancer drug docetaxel (DTX) and graphene quantum dot (GQD) nanocarriers in the presence and absence of peg-b-pla copolymer as surface modifier by molecular dynamics simulation. According to the results and analysis, the presence of the copolymer at the GQD interface first delayed the drug adsorption, but with the passage of time, the adsorption and solubility of DTX increased. This paper is well written, but needs the following revisions before publication:
- In the abstract, the author introduces the background too much, but introduces his work too little, which needs to be adjusted.
- In Figure 6, the text information is not obvious, and the author needs to make adjustments.
- What are the advantages of this job over other jobs? The author is advised to make a table for comparison.
- About “Carbon nanoparticles”, some relevant literature authors need to mention, such as: RSC Adv. 9, 41383-41391 (2019); Electrochim. Acta 168, 337–345 (2015); Talanta 134, 435–442 (2015); RSC Adv. 8, 42233–42245 (2018); Appl. Catal. A-Gen. 524, 163–172 (2016).
Author Response
- In the abstract, the author introduces the background too much, but introduces his work too little, which needs to be adjusted.
ANSWER: abstract revised regarding referees suggestion.
- In Figure 6, the text information is not obvious, and the author needs to make adjustments.
ANSWER: the quality of Fig 6 improved.
- What are the advantages of this job over other jobs? The author is advised to make a table for comparison.
ANSWER: As far as the authors know, no similar work has been done so far to make a comparison.
- About “Carbon nanoparticles”, some relevant literature authors need to mention, such as: RSC Adv. 9, 41383-41391 (2019); Electrochim. Acta 168, 337–345 (2015); Talanta 134, 435–442 (2015); RSC Adv. 8, 42233–42245 (2018); Appl. Catal. A-Gen. 524, 163–172 (2016).
ANSWER: all valuable references cited.
Round 2
Reviewer 1 Report
The manuscript was significantly improved, and I feel that it could be published essentially in the present, revised form apart from two small and very minor changes:
- Line 101: It is not the Leap-Forg, but the Leap-Frog algorithm. Please correct he misprint.
- Please include in the manuscript the polymer composition as given in the rebuttal letter: "In the simulated box we insert 10 polymer chains containing 7 PEG blocks and 3 PLA blocks." (blocks should however be replaced by repeat units).
Reviewer 2 Report
The article has been systematically modified and can be accepted.
Author Response
Thanks!